# Enhancing the In Vitro Biological Activity of Degraded Silk Sericin and Its Analog Metabolites

**DOI:** 10.3390/biom12020161

**Published:** 2022-01-19

**Authors:** Zhen-Zhen Wei, Yu-Jie Weng, Yu-Qing Zhang

**Affiliations:** School of Biology and Basic Medical Sciences, Medical College, Soochow University, RM702-2303, No. 199, Renai Road, Industrial Park, Suzhou 215123, China; 20204021006@stu.suda.edu.cn (Z.-Z.W.); 20184021002@stu.suda.edu.cn (Y.-J.W.)

**Keywords:** sericin, calcium hydroxide, ultrasonic degumming, antioxidant, lowering blood sugar

## Abstract

Two sericins of high and low molecular weight (HS and LS) were prepared from commercial silkworm cocoon silk with a boiling water and Ca(OH)_2_ solution with ultrasonic treatments, respectively. This study first investigated the release concentration of the two abovementioned sericins in simulated saliva, gastric juice, and intestinal juice (pH 6.8, 2.0, and 7.4, respectively) within 10 h. The results showed that the order of sericin release rate and its amount in the simulated environment was gastric juice > saliva > intestinal juice. Second, the molecular weights of both sericin metabolites formed by in vitro enzymatic degradation were lower than 15 kDa. The α-glucosidase inhibitory activities of both sericins and their analog metabolites were positively correlated with their concentrations. The IC_50_ values of the HS- and LS-derived metabolites were 1.02 ± 0.12 mg/mL and 0.91 ± 0.15 mg/mL, respectively, which were five to seven times lower than those of both original sericins. The total antioxidant capacities and hydroxyl radical scavenging capacities of both metabolites were enhanced by one- to three-fold compared with HS and LS. These results indicate that both sericins, regardless of molecular size, have significantly enhanced antioxidant, superoxide free radical scavenging, and glycosidase inhibitory activities after simulated metabolism, and that LS is better than HS regardless of simulated digestion. These results confirm that sericin is important in the sustainable development and utilization of silk resources, especially the reduction in environmental pollution, and provides new ideas for the development of adjuvant treatments for diabetes and the development of foods with anti-hyperglycemic functions.

## 1. Background

Since 2010, the annual output of silkworm cocoons in China alone has exceeded 600,000 tons, accounting for about 70% of the total global output. During the production of cocoon silk, about 40,000 to 50,000 tons of sericin globally are discharged in alkaline liquid waste every year. This not only causes serious environmental pollution and poor water quality, but also wastes a considerable amount of natural protein resources [1,2]. Therefore, cleaner processing technology and the functional development of sericin can not only lead to new uses for it, but more importantly, can reduce the level of environmental pollution caused by the degumming and degradation of this protein [3].

Sericin usually adheres to the outer layer of silk fiber. The main methods for separating and purifying sericin are high-temperature hydrolysis [4], high-temperature hydrolysis in an alkaline salt solution [5], acid extraction [6], protease degumming [7], neutral soap degumming [8], high-temperature and high-pressure degumming [9], and other degumming methods. Most commonly used in the laboratory are the solvent method, physical method, and biological enzymatic method. Alkaline solutions such as sodium hydroxide and sodium carbonate are the most commonly used solvents at present. However, it is difficult to separate these from sericin because the sodium salt formed later is easily soluble in water. The method developed in this laboratory using Ca(OH)_2_ aqueous solution as a degumming agent can completely remove sericin from silk fibers without destroying the mechanical properties of silk fibers. After being neutralized with acid, it can form insoluble precipitates. It can also form calcium sulfate precipitates upon reacting with sulfuric acid. Sericin can be used in plant fertilizers or building fillers without causing additional pollution [10]. Thus, this method of sericin separation is green and clean. We also found that the sericin obtained by the Ca(OH)_2_ degumming method has a lower molecular weight and higher in vitro antioxidant capacity, ultraviolet (UV) resistance, and inhibitory activity against α-glycosidase [11].

The large number of polar groups in sericin and their special structure result in it having several different biological properties. In different extraction processes, the peptide chain of sericin is broken to different degrees, producing different lengths of polypeptides, and thus exhibiting biological characteristics, such as antioxidant activity [12,13], the capacity to lower blood sugar [14], and anti-UV [15,16], anti-aging [17,18], antibacterial [19,20] and other in vitro properties. We used different methods to separate sericin, and obtained and compared the composition of outer sericin, inner sericin, and whole sericin in white commercial cocoons, and found that the three sericins contained serine (Ser), aspartic acid (Asn), and glycine (Gly), in quantities accounting for more than 50% of their total amino acids [21]. All three types of sericin exhibited obvious anti-UV activity. Among them, low-molecular-weight sericin peptides are more easily absorbed and utilized by the human body and have the potential to be used as raw materials for cosmetics. Amino acid analysis of an alcoholic extract of green cocoon sericin also led to similar conclusions. Ser, Asn, Gly, and proline (Pro) accounted for the largest proportion of amino acids in the alcoholic extract of green cocoon sericin, i.e., 46.98% of the total amino acid content. Testing with the 1,2′-azino-bis(3-ethylbenzthiazoline-6-sulphonic acid (ABTS) and FRAP methods showed that the antioxidant capacity of the four extracts was in the order: alcohol extract of green cocoon silk > outer sericin > whole sericin > inner sericin. In addition, the inhibition rate of outer-layered sericin (20 mg/mL) on *E. coli* could reach 50%. The IC_50_ values of tyrosinase inhibitory activity of outer sericin and whole sericin were 8.67 mg/mL and 3.49 mg/mL, respectively. The α-glucosidase inhibition rates of 10 mg/mL outer sericin and whole sericin reached 65.62% and 51.3%, respectively [22,23]. These results show that although these three sericins have some differences in physical and chemical properties, they all have obvious antioxidant, anti-UV, anti-bacterial, moisture absorption, and whitening properties, and can lower blood sugar levels in vitro. However, these results were all obtained via in vitro evaluation experiments. Thus far, there are no reports on in vitro evaluation experiments on simulated digestion products of degraded sericin. Therefore, this study simulated the digestion of high- and low-molecular-weight sericin in saliva, gastric juice, and intestinal fluid to explore the antioxidant properties, free radical scavenging ability, and in vitro antidiabetic activity of the digested products.

## 2. Materials and Methods

### 2.1. Materials

Commercial white cocoons (So-hao × Zhong-Ye) were purchased from Nantong New Silk Road Silk Co., Ltd. (Jiangsu, China); α-amylase (activity ≥ 50 U), pepsin (purity: 99%), trypsin (≥250 units/mg), the BCA protein content detection kit and other reagents were purchased from Sigma Co., Ltd.

### 2.2. Sericin Sample Preparation

An appropriate amount of cocoon shells was weighed, 90× distilled water (*W/V*) was added, and ultrasonic treatment was carried out at 100 °C for 2 h at an ultrasonic power of 600 W. The distilled water was replaced, and the above treatment was repeated once. The degumming liquid was collected and concentrated to an appropriate volume under negative pressure. The liquid was centrifuged at 10,000 rpm for 30 min to remove insoluble impurities, the supernatant was collected and concentrated again, and the concentrated solution was dried using a spray dryer to form a powdered sample with a higher molecular weight of sericin (HS).

A certain amount of cocoon shell was weighed and added to 0.025% (*W/V*) Ca(OH)_2_ solution at a bath ratio (1:90, *W/V*) [24]. Ultrasonic treatment was carried out at 100 °C for 2 h and boiling degumming was performed for 4 h at an ultrasonic power of 600 W. The Ca(OH)_2_ solution was replaced with fresh solution and the degumming was repeated once. The degumming liquid was collected and concentrated to an appropriate volume under negative pressure. The pH was adjusted to neutral with 1 mol/L sulfuric acid and the sample was left to stand for 12 h at 4 °C, then centrifuged at 10,000 rpm for 30 min to remove precipitated calcium sulfate and other insoluble impurities. The supernatant was collected and concentrated again. A powdered sample with a lower molecular weight of sericin (LS) was obtained using a spray dryer.

### 2.3. Preparation of Simulated Digestion Products

The two above-mentioned sericins (HS and LS) were diluted with distilled water to an appropriate concentration (10 mg/mL), and 18 mL of each was taken for later use. Then, 2 mL α-amylase was added to 18 mL sample solution and incubated at 37 °C for 10 min, following which the pH was adjusted to 2.0 ± 0.1 with 1 mol/L hydrochloric acid (three samples were taken, 1.0 mL for each sample, which was the A_10_ sample). Thereafter, 2 mL pepsin was added, the solution was incubated at 37 °C for 90 min, and the pH was adjusted to 7.0 ± 0.1 with 1.0 mol/L NaOH solution (three samples were taken, 1 mL each, which was the B_90_ sample). After adding 5 mL trypsin and 5 mL sodium cholate solution, the mixture was incubated at 37 °C for 90 min and the enzyme was inactivated using a water bath at 90–95 °C (three samples, 1.0 mL for each sample, were C_90_ samples). The concentrations of sericin in the HS and LS digestion products (EHS and ELS) were detected with the BCA detection kit.

### 2.4. In Vitro Simulated Sustained Release

The simulated digestion experiments of two sericin proteins were carried out according to the previously reported methods and procedures, with slight modifications [25,26]. HS and LS solutions each with a concentration of 10 mg/mL were put into a dialysis bag (molecular weight 1 kDa), sealed with a thin thread, and immersed in 10 mL of different buffers containing Tween 80. The three buffers of different pH value (2.0, 6.8, and 7.4) simulated gastric juice, intestinal juice, and blood, respectively. The samples were fixed in a constant temperature shaker at 37 °C and rotated at a speed of 100 rpm for 0.5, 1, 2, 4, 6, 8, 10, and 12 h. The same volume of buffer solution was a control, the BCA method was used to determine the total protein content in the dialysis bag sustained-release sample, and the slow-release curves of the two sericins were plotted. The experiment was set up in three parallel groups. These sustained-release sericin samples simulated in vitro are referred to as EHS or ELS.

### 2.5. Determination of Molecular Weight Range

The molecular weight range of four proteins (HS, LS, EHS, and ELS) was determined by SDS-PAGE, according to the method previously reported by the author [27].

### 2.6. Determination of α-Glucosidase Inhibitory Activity

HS, LS, EHS, and ELS samples were prepared as protein solutions of a given concentration. The α-glucosidase inhibitory activity was determined by enzymatic reaction according to the method previously reported by the author [28].

### 2.7. Determination of α-Amylase Inhibitory Activity

Precisely 40 μL of the above four sample solutions was taken in tubes, 40 μL of 0.02 U/mL amylase was added, and samples were incubated in a 37 °C water bath for 10 min. Once again, 40 μL amylose solution was added and incubated in a 37 °C water bath for 10 min, following which 20 μL of 1 mol/L hydrochloric acid was added to stop the reaction. Then, 50 μL colorimetric iodine solution was added and the absorbance at 620 nm was measured and recorded as A. With PBS as a blank control, the above steps were repeated. The absorbance value at 620 nm was A_0_, and the experiment was conducted on three groups in parallel. Calculation formula: α-amylase inhibitory activity (%) = (1 − A/A_0_) × 100.

### 2.8. Determination of ABTS and Hydroxyl Radical Scavenging Rate

The ABTS method used in this study is an indirect method to detect the antioxidant capacity of substances. Oxidized ABTS is a stable blue–green cationic ABTS^+^ radical, soluble in acidic ethanol or aqueous phase, with an absorption maximum detected at 734 nm. After ABTS^+^ was added to the solution to be tested, the antioxidant component reacted with ABTS^+^ and then faded. At this time, the change in absorbance at 734 nm was detected. Trolox was used as a control system to quantify the antioxidant capacity of the solution to be tested [29]. The scavenging activity of HO· was determined according to a previously described method [30].

### 2.9. Statistics

The experimental data are expressed as the mean ± standard deviation (SD). Significant differences between two sets of data were assessed using one-way ANOVA (Origin version 8.5). A *p*-value < 0.05 was considered statistically significant.

## 3. Results

### 3.1. In Vitro Simulated Sustained Release

In order to investigate the drug accumulation effect of sericin in different organs, this experiment utilized the dialysis bag method to determine the cumulative concentrations of two sericins (HS and LS) in simulated saliva, gastric juice, and intestinal juice (pH 6.8, 2.0, and 7.4, respectively) within 12 h. As shown in Figure 1, the sericin accumulation rate is related to the types of sericin and digestive juice. Regarding the sericin type, the accumulation speeds and concentrations of LS in the three simulated digestive juices were higher than those of HS. In comparison with the cumulative effect of both sericins in different simulated digestive juices, the accumulation speed and amount of sericin in simulated gastric juice were the highest, followed by simulated saliva and then simulated intestinal juice.

### 3.2. Molecular Weight Range of Sericin

In order to explore the differences in the molecular weight of sericin obtained by different degumming methods before and after simulated digestion in vitro, the molecular weight ranges of four sericins were determined by SDS-PAGE. As shown in Figure 2, the molecular weight range of the four powdered sericins was ordered as follows: ELS < EHS < LS < HS. HS had the largest range of molecular weight and a relatively concentrated distribution. The electrophoresis bands were mainly concentrated in the 15–50 kDa range, indicating that the degradation effect of boiling water combined with ultrasound on sericin is relatively small. The band color of LS was relatively light, but its molecular weight distribution was the widest, which indicates that most sericin peptides were degraded into smaller peptides and amino acids. The molecular weight distribution of sericin peptides obtained by silk degumming in 0.025% calcium hydroxide aqueous solution was concentrated below 20 kDa [22]. It also indicates that the 0.025% Ca(OH)_2_ solution has a better degradation effect on sericin. The molecular weight distribution range of EHS was much smaller than that of HS, and the electrophoresis bands were mainly concentrated below 15 kDa. The color of the ELS bands was lightest, and it mainly concentrated below 15 kDa, which also shows that its range of molecular weight is the smallest. EHS and ELS showed no obvious traces in the molecular weight range of 15–50 kDa. This shows that HS and LS are effectively decomposed into low-molecular-weight peptides and hydrolates after simulated digestion in vitro.

### 3.3. α-Glucosidase Inhibitory Activity

α-Glucosidase can dissociate polysaccharides into monosaccharides, which mainly exist in the mucosa of the small intestine, and is essential for the sugar metabolism pathway. If its activity is inhibited, the catabolism of carbohydrates is inhibited, thereby reducing blood sugar levels. After the experimental data was fitted and analyzed by the software Origin 8.5, as shown in Figure 3a, in the range of 1.0–10 mg/mL, the concentration of HS and its ability to inhibit the activity of α-glucosidase were dose-dependent, and its IC_50_ was 5.64 ± 0.22 mg/mL, a relatively low concentration of HS showed a good inhibitory effect on α-glucosidase activity. As shown in Figure 3b, LS concentration and α-glucosidase activity inhibited IC_50_ was 4.83 ± 0.23 mg/mL. After HS was subjected to mock digestion, the IC_50_ of EHS concentration and glucosidase inhibitory activity was 1.02 ± 0.12 mg/mL. IC_50_ of ELS concentration and glucosidase inhibitory activity was 0.91 ± 0.15 mg/mL. After simulated metabolism in vitro, both HS and LS could more effectively inhibit the degradation of carbohydrates, which is expected to reduce the blood glucose level in the body.

### 3.4. α-Amylase Inhibitory Activity

α-Amylase can attack the α-1,4-glycosidic bond of starch to form reducing sugars, mainly trisaccharides; thus, it can reduce the viscosity of starch and is also called a liquefaction enzyme. After fitting and analyzing the experimental data by the software Origin 8.5, as shown in Figure 4a, in the range of 0.1–1 mg/mL, the concentration of HS was dose-dependent with the inhibition of α-amylase. When the concentration was 1.05 mg/mL, the highest inhibition rate was 40.65%, and the IC_50_ between EHS concentration and α-amylase inhibitory activity was 0.37 ± 0.02 mg/m. Relatively low concentrations of HS showed a better inhibitory effect on α-amylase activity. As shown in Figure 4b, when the LS concentration was 1 mg/mL, the highest inhibition rate was 52.57%, and the IC_50_ of ELS concentration and α-amylase inhibitory activity was 0.28 ± 0.05 mg/mL. The above results indicate that after simulated metabolism in vitro, both HS and LS can more effectively inhibit starch hydrolysis, thereby exerting a hypoglycemic effect in the body.

### 3.5. Total Antioxidant Capacity

ABTS^+^ is blue–green in color. Antioxidant substances react with ABTS^+^ to make solutions fade; therefore, the ABTS scavenging ability can be expressed by the absorbance value. In this study, the ABTS scavenging abilities of two sericins (HS and LS) at concentrations of 1–10 mg/mL, and their degradation products (EHS and ELS) at 0.1–1.0 mg/mL, were determined. After fitting and analyzing the experimental data by software Origin 8.5, as shown in Figure 5, the scavenging ability of ABTS increased with the increase in sample concentration. The IC_50_ of HS concentration and ABTS clearance was 8.64 ± 0.12 mg/mL. The IC_50_ of LS concentration and ABTS clearance was 6.79 ± 0.07 mg/mL. The IC_50_ of EHS concentration and ABTS clearance was 2.48 ± 0.06 mg/mL. The ABTS clearance IC_50_ of ELS was 2.72 ± 0.04 mg/mL. The experimental results showed that before the simulated digestion, the ABTS scavenging ability of LS was significantly higher than that of HS. After the simulated digestion, there were no significant differences in the ABTS scavenging abilities between the two products.

### 3.6. Hydroxyl Radical Scavenging Capacity

Hydroxyl free radicals (**⋅**OH) are the most active and toxic free radicals and can cause damage to cells. In vitro detection is mainly carried out with the Fenton reaction. Hydroxyl radicals react with salicylic acid to produce dihydroxybenzoic acid, which has an absorption peak at 510 nm. When the components in a sample react with hydroxyl radicals, a change in absorbance is induced. Figure 6 shows that when the sericin concentrations were 1 mg/mL and 2 mg/mL, the hydroxyl radical scavenging rate remained below 25%, and when the concentration reached 6 mg/mL, the hydroxyl free radical scavenging ability was significantly enhanced. After fitting and analyzing the data by Origin software, the IC50 of the hydroxyl radical scavenging rate of HS, LS, EHS and ELS were, respectively, 7.49 ± 0.12 mg/mL, 6.33 ± 0.12 mg/mL, 4.53 ± 0.12 mg/mL, 3.25 ± 0.12 mg/mL, 0.12 mg/mL. This figure shows that, regardless of the size of sericin molecules, they have a strong ability to scavenge hydroxyl radicals, their ability is stronger after digestion, and there is no significant difference between the two, whether they are digested or not.

## 4. Discussion

In this study, a clean Ca(OH)_2_-ultrasonic method was used to obtain LS and a green high-temperature method was used to obtain HS. Researchers in the laboratory have reported the degumming rate, molecular weight range, and amino acid composition of LS; the degumming rate of the Ca(OH)_2_-ultrasonic method can reach 28.6% [22]. At the same time, the physical and chemical properties and in vitro biological activity of this low-molecular-weight sericin peptide and its hydrolysate were also investigated and analyzed in detail [23]. The results show that the concentration of sericin that inhibits α-glucosidase activity by 51.3% in vitro is 10 mg/mL, which is also very close to the results of this experiment. In this experiment, the IC_50_ value of LS inhibition of α-glucosidase activity was 6.83 ± 0.23 mg/mL. The IC_50_ value of the outer layer of HS prepared by the boiling method was slightly lower, at 5.64 ± 0.22 mg/mL. The IC_50_ values of their metabolites, ELS and EHS, were 0.91 ± 0.15 mg/mL and 1.02 ± 0.12 mg/mL, respectively, and the IC_50_ values were significantly reduced by about five-fold. The α-amylase inhibitory activities of LS and HS sericin were still relatively high. When the concentrations of both reached 1.0 mg/mL, their inhibition rates reached 40–45%. After the simulated digestion, the inhibitory activity of the product also increased seven-fold, and the IC_50_ values of the α-amylase inhibitory activity reached 0.2864 ± 0.05 mg/mL and 0.371 ± 0.02 mg/mL, respectively. This shows that after simulated metabolism, whether it is HS or LS and their metabolites, their hypoglycemic activities in vitro are greatly enhanced. Sericin can reduce the solubility of cholesterol in solution. In vitro cell experiments found that sericin can reduce the uptake and utilization of cholesterol by intestinal cells [31]. Determining the uptake of radiolabeled cholesterol in sericin solution in human cloned colon adenocarcinoma cells (Caco-2), Limpeanchob et al. found that sericin levels as low as 25 and 50 μg/mL inhibited the entry of cholesterol into Caco-2 by 30% At the same time, it was found that administering sericin can reduce the levels of high density lipoprotein (HDL) and cholesterol (CHOL) in rats with high cholesterol diet, i.e., sericin regulates blood lipid levels in the body by reducing the utilization of cholesterol by intestinal cells and enabling the precipitation of cholesterol [32]. This result is also consistent with the results of our recent hypoglycemic test in diabetic mice and rats using this small-molecular-weight sericin peptide and its hydrolysate [14,33]. After 4 weeks of feeding, this degraded sericin peptide and its hydrolysate could significantly improve the physiological and biochemical indicators of STZ-induced diabetic mice and rats, the abnormal glucose tolerance and insulin tolerance were significantly improved, the serum insulin level was reduced, and insulin resistance decreased whereas insulin sensitivity increased. The content of glycated serum protein (GSP) in serum was significantly reduced, and fasting blood glucose was significantly reduced or close to the levels in normal mice, which can improve the activity of serum liver function enzymes and reduce the levels of CHO1, TG (thyroglobulin), HDL, LDL (low-density lipoprotein) and LDH (lactate dehydrogenase), thereby improving blood lipid metabolism. Immunofluorescence staining showed that consuming sericin can improve the abnormal secretion of pancreatic insulin in diabetic rats. Therefore, the oral tests in diabetic mice and rats in vivo and the biological activity tests of digestion products simulated in vitro both showed that LS obtained by calcium hydroxide degumming is digested and absorbed by the stomach and intestines. The smaller molecules of the sericin peptide and its hydrolysates enter the blood to reach various organs, especially the liver. The enhancement of the body’s antioxidant effect could reduce the body’s oxidative stress response and inflammation, leading to increased serum antioxidant enzymes, improved lipid metabolism, glucose metabolism, and insulin metabolism, finally tending to return blood sugar to normal levels. The above results fully indicate that the antioxidant capacity and ability to inhibit free radicals of sericin and its metabolites in mice contribute significantly to lowering blood sugar levels, whereas their in vitro hypoglycemic activity plays a minor role in lowering blood sugar levels in vivo.

## 5. Conclusions

In this study, both high-temperature and Ca(OH)_2_-ultrasonic methods were used for degumming the cocoon shells of silkworms to cleanly recover sericin protein. HS and LS (sericins of high and low molecular weight, respectively) were obtained. The two kinds of sericin were processed by in vitro simulated digestion and two metabolites, EHS and ELS, were obtained. By comparing the molecular weight range of sericin and its metabolites by SDS-PAGE, it was found that after simulated digestion, the molecular weights of the two metabolites decreased significantly, indicating that both sericins were efficiently degraded. At the same time, the biological activity of sericin and its metabolites was measured. Compared with HS and LS, the antioxidant activities and glucosidase inhibitory activities of EHS and ELS were significantly enhanced, indicating that HS and LS may be hydrolyzed into smaller molecules, polypeptides, or oligomeric peptides in the body. Regardless of their molecular size, sericins could eliminate free radicals in the body. They could also reduce oxidative stress, inhibit the degradation of carbohydrates by hydrolase, and reduce the damage to the body caused by high sugar levels. In conclusion, the experimental results in this article show that small-molecular-weight sericin with better water solubility has better development potential in functional foods, which can help to lower blood sugar. The next step in our laboratory is to feed the small-molecule sericin and hydrolysate recovered from silk processing as biological drugs to type 2 diabetic rats to explore their hypoglycemic effect and mechanism.

## Figures and Tables

**Figure 1 biomolecules-12-00161-f001:**
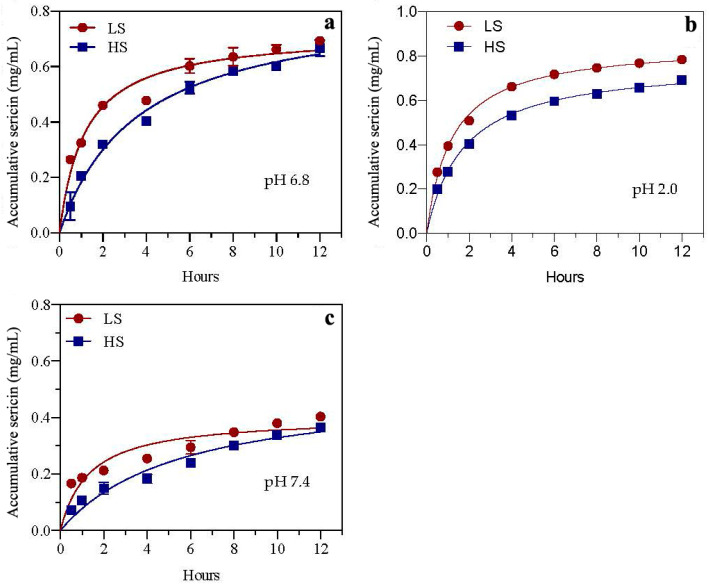
Drug accumulation rates of two sericin peptides in simulated digestive juices pH 6.8 (**a**), pH 2.0 (**b**) and pH 7.4 (**c**) in vitro. HS and LS: higher and lower molecular weight sericin, respectively; EHS and ELS: enzymatic digestion products of HS and LS, respectively.

**Figure 2 biomolecules-12-00161-f002:**
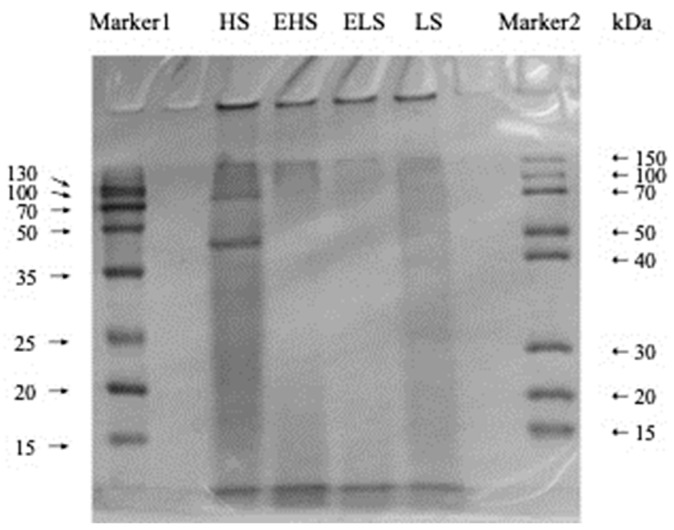
SDS-PAGE (15% gel) of sericin peptides and their digestive metabolites. HS and LS: higher and lower molecular weight sericin, respectively; EHS and ELS: enzymatic digestion products of HS and LS, respectively.

**Figure 3 biomolecules-12-00161-f003:**
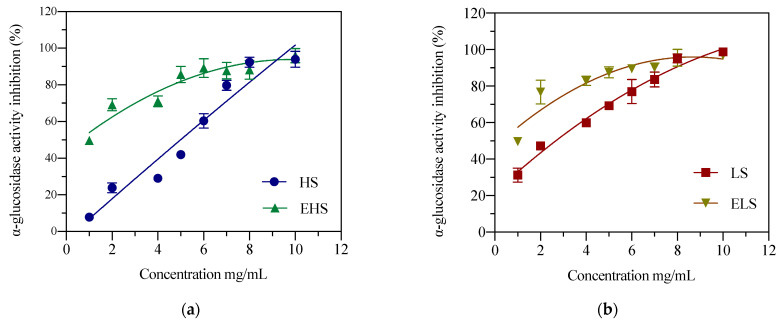
The inhibition activities of both HS (**a**) and LS (**b**) and their enzymatic metabolites on α-glucosidase. HS and LS: higher and lower molecular weight sericin, respectively; EHS and ELS: enzymatic digestion products of HS and LS, respectively.

**Figure 4 biomolecules-12-00161-f004:**
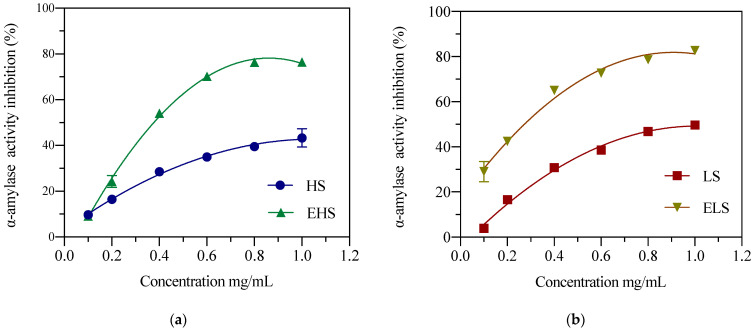
The inhibition activities of both HS (**a**) and LS (**b**) and their enzymatic metabolites on α-amylase. HS and LS: higher and lower molecular weight sericin, respectively; EHS and ELS: enzymatic digestion products of HS and LS, respectively.

**Figure 5 biomolecules-12-00161-f005:**
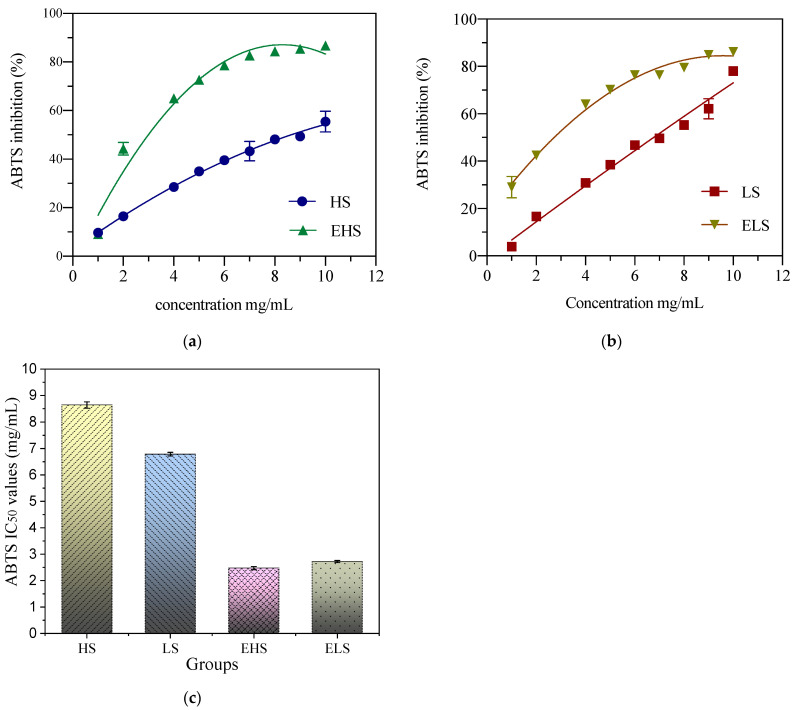
The ABTS scavenging ability of both peptides HS (**a**) and LS (**b**) and their enzymatic metabolites (**c**). HS and LS: higher and lower molecular weight sericin, respectively; EHS and ELS: enzymatic digestion products of HS and LS, respectively.

**Figure 6 biomolecules-12-00161-f006:**
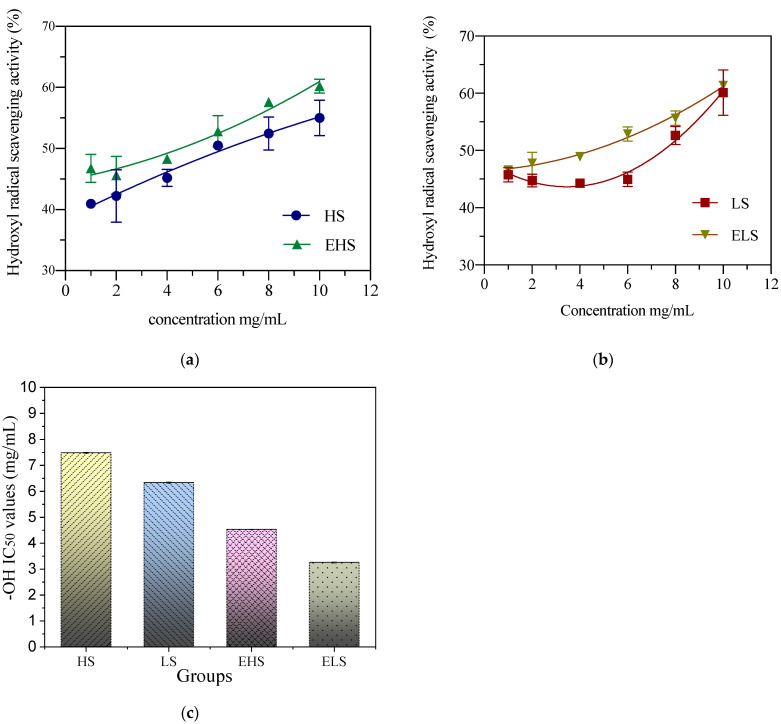
The hydroxyl radical scavenging activity of both sericins HS (**a**) and LS (**b**) and their digestive metabolites (**c**). HS and LS: higher and lower molecular weight sericin, respectively; EHS and ELS: enzymatic digestion products of HS and LS, respectively.

## Data Availability

Code and material; The datasets used and/or analyzed during the current study as well as analysis scripts are available from the corresponding author on reasonable request.

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
