# Peer review of "Enhancing the In Vitro Biological Activity of Degraded Silk Sericin and Its Analog Metabolites"

_biomolecules, 2022, doi:10.3390/biom12020161_

Round 1

Reviewer 1 Report

The article - The degraded silk sericin and its analog metabolites and enhanced in vitro biological activity -presents interesting results on antioxydant and antidiabetic potential of sericins and their metabolites formed after the in vitro simulation of a gastrointestinal digestion. However some points must be modified to improve the quality of the document :

P1, line 2 : the title is not clear, probably because of the « and » between metabolites and enhanced, couls the authors improve the title ?

P2, line 64 to 73 : the font size is different from the rest of the document

P3, line 106 : Could the authors specify the composition of these extracts in sericins ? Are they not rather enriched fractions ?

P3, line 118 : I am a little surprised not to see any references for the preparation of the solutions as well as for the in vitro digestion protocol ? There are reference or consensus protocols. In this context, can the authors explain their choice ? The hydrolysis times, for example, do not seem to correspond to a physiological reality, or in which animal ?

P4, line 157 : Can the authors specify the cut-off threshold for dialysis bags and how this step can simulate absorption? Is this step carried out before each biological activity test?

P4, Figure 1 : Can the authors explain how to measure the sericin concentration in the different compartments? In my opinion the legend deserves to be developed for a better understanding of the figure. This last remark can be extended to all the figures ....

P5, line 183 : I agree with the authors, it would probably be useful to make a gel more suitable for the study of fragments resulting from enzymatic digestions. I'm not sure that the presented gel gives us a lot of information ....

P8, line 280 : Can the authors confirm that there are no other refrences of work on sericins and their biological activities in vitro, because the discussion really only mentions studies carried out by the same laboratory (or by the sweaters themselves?)

In general, the preliminary results obtained in this article are of real interest in an objective of valuing industrial co-products towards the functional food or nutraceutical market. However, some elements are missing, such as the yield of each extraction for example, to make this article really convincing. I think the authors can support their argument. The quality of English also deserves to be reviewed by a native spoken.

Reviewer 2 Report

The Introduction is quite well presented, but some improvements are needed.

Line 32: 40-50,000 tons of sericin, is it worldwide or in China?

Lines 56-60 references are missing

Lines 73-82 In the first part references are missing. Moreover, the novelty of this paper should be better explained. What do you mean with “rarely related”? Which is the actual knowledge on this topic?

Paragraph 2.1 Which was the purity of the enzymes?

Paragraphs 2.5, 2.6 and 2.8: Please summarize the methods briefly

Paragraph 3.3/3.4/3.5/3.6: What is the use of these regression equations? Are you sure they are noteworthy? What is the physical meaning of the coefficients of the equations? Why do the coefficients have different number of significant digits?

Paragraph 4: The discussion of the results could be improved. In particular, you should cite other works on sericin in the literature and discuss the differences in results. A comparison of the efficiencies of the degumming methods would also be interesting. Moreover, in the second part of the paragraph, the citations are missing. The meaning of the acronyms used in this paragraph is missing, as it is the only time they are mentioned.

The conclusions are to be completely modified, since they must not only be a summary of the results obtained. What could be the practical interest of your findings? What are the next necessary studies in this field?

Minor remarks

Line 9 were instead of are

Line 10: do not repeat boiling

Lines 15-18 Not clear, revise English

Lines 33-34 Not clear, revise English

Lines 34-37: do not repeat sericin too much

Lines 46-49 Not clear, revise English

Line 61 not clear the usefulness of obtained

Lines 64-73 check the size of the text

Line 89 what is 90X? 90%?

Line 101 (and elsewhere) Use M instead of mol/L

Lines 124-127 Not clear, revise English

Figure 1 avoid / before mg/mL

Line 203 Point is missing at the end

Lines 254-257 Not clear, revise English

Round 2

Reviewer 2 Report

I thank the authors for their answers.

The paper still needs some minor revisions:

  • In lines 81/88 there are some repetitions in the objectives of the paper. Moreover, the novelty is still not clear. What do you mean with " There are little in vitro evaluation experiments on degraded sericin especially its simulated digestion products "? Which are the published papers that evaluated the simulated digestion of degraded sericin? Which is the difference with these papers?
  •  Paragraph 3.3/3.4/3.5/3.6: The regression equations are useful only for the calculation of IC50. I repeat my questions. Are you sure they are noteworthy? What is the physical meaning of the coefficients of the equations? Why do the coefficients have different number of significant digits? If you don't explain this, I don't see why to put the equations in the results.
